# Intensive longitudinal follow-up of cisgender and transgender women engaged in sex work during the three months following initiation of daily oral PrEP: A series of case-studies with mixed-method assessments

Laia J. Vazquez Guillamet[1,2]*, Jorge Valencia[3,4], Pablo Ryan[5,6], Mariano Matarranz[5], Guillermo Cuevas-Tascón[5], Miguel Angel del-Olmo-Morales[7], Laura Laguna[8], Natalia Casanueva[9], Cristina Viladomiu[10], Lynn T. Matthews[11], Jeffrey V. Lazarus[1,2,12], Guillaume Chevance[13,1]*

1 Barcelona Institute for Global Health (ISGlobal), Hospital Clínic, University of Barcelona, Barcelona, Spain, 2 Facultat de Medicina i Ciencies de la Salut, Universitat de Barcelona, Barcelona, Spain, 3 Fundación de Investigación Biomédica, Hospital Infanta Leonor, Madrid, Spain, 4 Madrid Positivo Non-Governmental Organization, Madrid, Spain, 5 Departamento de Medicina Interna, Hospital Infanta Leonor, Madrid, Spain, 6 Centro de Investigación Biomédica en Red en Enfermedades Infecciosas (CIBERINFEC), Madrid, Spain, 7 Asociación IN GENERO (Interculturalidad y Género), Ciudad Real, Spain, 8 Departamento de Farmacia, Hospital Infanta Leonor, Madrid, Spain, 9 Facultad de Medicina, Universidad Complutense de Madrid, Madrid, Spain, 10 Independent Researcher, Barcelona, Spain, 11 Department of Medicine, Division of Infectious Diseases, University of Alabama at Birmingham, Birmingham, Alabama, United States of America, 12 CUNY Graduate School of Public Health and Health Policy, New York, New York, United States of America, 13 Univ Rennes, Inserm, EHESP, Irset (Institut de Recherche en Santé, Environnement et Travail) - UMR_S, Rennes, France

* lvazquezguillamet@uabmc.edu (LJVG); guillaume.chevance@ehesp.fr (GC)

## Abstract

Pre-exposure prophylaxis (PrEP) usage among cisgender and transgender women sex workers in Europe is low. This mixed-methods study examined the daily experiences of woman sex workers using PrEP, with emphasis on understanding the dynamic process of initiating and sustaining PrEP adherence. We employed an intensive longitudinal design with daily assessments of self-reported use of daily oral PrEP, side-effects, condom use, and number of clients over a 3-month period, followed by in-depth exit qualitative interviews that also explored PrEP initiation, communication with sex workers and clients, and stigma. Convenience sampling was used to enroll 15 sex workers (12 transgender and 3 cisgender women) presenting to a PrEP clinic in Madrid between November 2022 and January 2023, all from Latin America. We collected 1266 daily survey responses and 13 interviews. Quantitative results showed that average PrEP adherence was above 70%, with missed doses being randomly distributed, and that condom use and number of clients did not significantly change during the study period. Qualitative results showed that the main reason to start PrEP was concern about condom rupture and/or removal by clients. Facilitators of PrEP adherence included personal motivation, creation

**Data availability statement:** The data that support the findings of this study are openly available in Open Science Framework (https://osf.io/6rdxh/), this includes questionnaires, dataset, statistical code, interview guide, qualitative codebook, and thematic summaries. Full interview transcripts have not been shared to protect participants' identities.

**Funding:** This work was supported by "la Caixa" Foundation (ID 100010434, fellowship code LCF/BQ/DI20/11780017, awarded to LJVG), the European Union ("NextGenerationEU"/ "PRTR", awarded to GC), the Spanish government (grant RYC2021-033537-I, supported by MCIN/AEI/10.13039/501100011033, awarded to GC), and the Ministry of Research and Universities of the Government of Catalonia (2021 SGR 01563, awarded to GC). This work was supported by the Spanish government (grant CEX2023-0001290-S, supported by MCIN/AEI/10.13039/501100011033) and the Generalitat de Catalunya (through the CERCA Program). The funders had no role in study design, data collection and analysis, decision to publish, or preparation of the manuscript.

**Competing interests:** The authors have declared that no competing interests exist.

of daily routines, and use of personal alarms. Barriers to PrEP adherence included side effects from PrEP and its coingestion with other substances, and changes in daily routine related to work, and travel. Strict prescription protocols represented an additional layer of difficulty. Secondary gains included a feeling of empowerment and the opportunity to opt for condom substitution for economic benefit, personal pleasure, or the desire to foster a trusting relationship with long-standing clients. This involved few selected encounters and did not impact overall number of clients. PrEP communication was limited by PrEP and HIV stigma. We found a complex interplay of individual, occupational, and structural factors shaping early PrEP adherence among participants.

## Introduction

In 2023, Spain reported an average of 7.3 new HIV cases per 100,000 inhabitants [1]), nearly twice the European average of 3.8 [2], with the Community of Madrid consistently recording the highest number of new HIV diagnoses nationwide [1]. To curb the epidemic, in 2019 Spain introduced fully subsidized pre-exposure prophylaxis (PrEP) for individuals at high risk, including sex workers. PrEP is available as a daily pill (245 mg of tenofovir disoproxil fumarate and 200 mg of emtricitabine) and delivered through authorized healthcare centers, mostly hospital-based infectious diseases clinics, alongside other preventive measures [3].

Although sex work itself is not illegal in Spain, exploitation and pimping are prohibited. Estimates suggest there are approximately 114,576 women sex workers in Spain [4], of whom approximately 7% are transgender women (individuals assigned male at birth who identify as women) [5,6]. HIV prevalence among sex workers is estimated to be 0.8% to 2% among cisgender women (individuals assigned female at birth who identify as women) [5], higher among intravenous drug users and Spanish nationals, and 25% to 27% among transgender women [7–10]. Despite being a key population, women sex workers in 2024 represented only 0.7% of PrEP users in Spain (gender is not accounted for in this statistic) [11].

As in other countries, current evidence indicates that the main barrier to PrEP initiation among women sex workers in Spain is lack of awareness, particularly among cisgender women [12,13]. How women sex workers learn about PrEP and what motivates them to start PrEP remains poorly documented. Once initiated, adherence of four or more doses per week is considered sufficient to provide optimal HIV protection among cisgender and transgender women [14,15]. Nevertheless, several studies have shown a decrease in PrEP adherence over time, particularly among sex workers, cisgender, and transgender women, which increases the risk of HIV infection and disengagement from care [15–21]. Known barriers to PrEP adherence include PrEP- and sex-work-related stigma, difficulties managing side effects, costs, migratory status, problematic drug use, and survival priorities [12,17,20,22–29]. We have yet to understand how these barriers translate into the day-to-day experience of women sex workers taking PrEP and which behaviors they engage in to ensure consistent use.

Importantly, PrEP use has been associated with a decrease in condom use and/or an increase in the number of sexual partners [30,31]. These behavioral compensations are relevant because they may lead to an increase in sexually transmitted infections (STIs) other than HIV [30,31], and to unplanned pregnancies. To date, two prospective studies conducted in Benin and Thailand among cisgender and transgender women sex workers, respectively, have found no evidence of reduced condom use following PrEP initiation and have documented declines in STI diagnoses over the follow-up period. Concerns also exist that sex workers may be pressured into condomless sex if clients are aware that they take PrEP [24], or due to market pressure (for example, if prices are driven down because of PrEP use) [32]. Yet little is known about PrEP communication among sex workers and clients. Learning about changes in condom use, the number of clients, and communication patterns is important to understand their relationship with PrEP adherence and to inform multifaceted HIV prevention programs that address PrEP-associated behavioral spillovers through appropriate messaging, STI care, family planning services, and empowerment.

Originally developed to guide clinicians in evaluating adherence among patients with hypertension, the Medication Adherence Model (MAM) provides a useful framework for understanding how individuals make and sustain medication-taking decisions. Through its three core concepts (Purposeful Action, Patterned Behavior and Feedback), the model provides a structured lens for examining what motivates individuals to take or skip medication, how daily patterns form over time, and how these patterns interact with broader structural and occupational contexts such as sex work. Moreover, MAM's distinction between intentional nonadherence (e.g., cycling off during perceived low-risk periods) and unintentional nonadherence (e.g., forgetting doses or facing access barriers) is particularly relevant for providers, as it highlights different points of intervention to support adherence. Guided by this framework, we conducted an intensive longitudinal study to better understand the phenomenon of PrEP adherence and its bidirectional relationship with sex work. Self-reported PrEP adherence, side effects, condom use, and number of clients were monitored daily for three months via a smartphone app among women sex workers in Madrid, Spain, after initiating PrEP. This was followed by an in-depth explanatory qualitative interview that also explored PrEP initiation, PrEP communication with clients and coworkers, and PrEP-related stigma. This study aims to provide an in-depth, daily account of the lived experiences of women sex workers using PrEP, generating insights that can inform future interventions.

## Materials and methods

### Ethics statement

The protocol was reviewed and approved by the Ethics and Research Commission from the Hospital Universitario Infanta Leonor and the Hospital Virgen de la Torre (086–22) in June 2022. Formal written consent was obtained from all participants.

### Study population, context and inclusion criteria

Most available data about sex work in Spain come from women accessing outreach services, which are predominantly independent workers, migrants from Latin America (about half with regularized status), and 35–7% street-based [6,8,10,33]. Spanish-born women, who represent 5–10% in more recent cohorts [6,7,10], are disproportionately represented among those affected by substance dependence and homelessness, particularly among cisgender sex workers. [12,33] Condom use with clients among cohorts ranges from 75% to 100% [10,12,13,34–36]. Inconsistent use among cisgender women is linked to being Spanish-born, substance dependence, violence, prior condom rupture, and a history of STI [10,34,35,37]; while among transgender women it is more often reported in private sporadic encounters [8]. In both groups, condom rupture and coercion with clients are common even among those reporting consistent use [8,34]. Cocaine is widely consumed in professional contexts due to client demand, while transgender women additionally report use of Viagra, poppers, and less frequently substances such as GHB, MDMA, or methamphetamine [8]. Transgender women are also reported to engage in national and international work-related travel [8]. Stigma, violence, and mental health issues are widespread in both communities, with transgender women facing added barriers due to transphobia [8,38].

Within the national healthcare system, women sex workers are eligible for PrEP if they disclose their sex work status and report inconsistent condom use [11]. Both transgender and cisgender women sex workers may also qualify without disclosure of sex work if they report at least two high-risk behaviors in the past year, with cisgender women additionally required to report inconsistent condom use. High-risk behaviors include having more than ten sexual partners, condomless anal sex, drug use associated with unprotected sex, repeated use of post-exposure prophylaxis, or a diagnosis of a bacterial STI [11].

Building on these national criteria, potential study participants were approached during their initial PrEP visit at the Hospital Infanta Leonor in Madrid, Spain. Sex workers were invited to participate in the study if they were: (1) cisgender or transgender women; (2) sex workers; (3) 18 years or older; (4) fluent in Spanish; (5) meeting medical criteria to start PrEP immediately (normal kidney function and no medical conditions requiring additional monitoring, such as chronic hepatitis B infection); (6) able to attend medical visits; (7) and owners of a personal mobile phone with internet access allowing the installation of a smartphone app. For this study, sex work was defined as the exchange of sex for money or goods.

## Study design and data collection

Participants were screened and consecutively enrolled in the order they presented to a suburban hospital-based PrEP clinic during a fixed recruitment period from November 8, 2022 to January 30, 2023. The non-governmental organizations (NGOs) Madrid Positivo and In Género, collaborators in prior projects, informed women sex workers in their networks about the clinic and provided practical guidance on how to attend. Some participants were already familiar with PrEP and the clinic's services through earlier fieldwork surveys assessing interest in PrEP [12,13]. Peer referral later expanded participation beyond the initial NGO network.

The initial PrEP clinic visit routinely includes assessment of risk factors for HIV infection that qualify individuals for fully subsidized PrEP under the national health system, such as engagement in sex work; this information is subsequently submitted to the national surveillance program for PrEP use [11]. During the same visit, individuals undergo evaluation of kidney function and screening for HIV, viral hepatitis, and other STI. They also receive information about the schedule of follow-up appointments, which encompass medical evaluations, laboratory testing, and PrEP dispensation at the hospital pharmacy. In the context of this study, all women enrolled self-identified as sex workers at the initial visit and confirmed that they had the means and time to attend the scheduled appointments.

For those women interested in participating in the study and meeting inclusion criteria, written informed consent was obtained after their first medical PrEP visit, followed by a brief demographic questionnaire. After lab work results for kidney function and hepatitis B virus exposure confirmed participants' readiness to start PrEP, they were contacted through their personal phone number and invited to download the *m-path* application on their phones [39], through which they received brief daily self-reported questionnaires. M-path is a GDPR (General Data Protection Regulation) compliant application developed by the Faculty of Psychology and Educational Sciences at KU Leuven for mobile assessment in behavioral research. It was not specifically designed for this study.

After completing the three months of daily phone questionnaires, participants scheduled an in-depth exit qualitative interview at a private location of their choosing. If unable to meet in person after several attempts, the interviews were conducted by phone. Interviews were carried out by trained personnel (NC, LL, LVG). At the time of the interview both the interviewer and the interviewee had access to a graphic with the results of the participant's daily phone questionnaires. Interviews lasted approximately 30–60 minutes and were audio recorded and transcribed. Participants received 10 euros for each of the three study appointments (screening, enrollment, exit interview), as well as 0.5 euros for each daily survey completed, representing a maximum of 75 euros.

## Quantitative measures

PrEP adherence, side effects, number of clients, and condom use were measured every day with a single item for each variable detailed below (screenshots of the app-based questionnaire are shown in Fig 1). We selected these variables based on their relevance to the study outcome (understanding the phenomenon of PrEP adherence and its bidirectional relationship with sex work), and their suitability for daily measurement, which allowed us to capture short-term fluctuations and day-to-day patterns in participants' experiences.

**PrEP adherence.** Daily PrEP adherence was measured with the following item: "Did you take the PrEP pill yesterday?" to which participants replied Yes or No.

**PrEP-related side effects.** Side effects were measured by asking participants the following question: "Please indicate below if you had any adverse effects yesterday due to PrEP (you can choose more than one option): nausea/vomiting, lack of appetite, abdominal pain, headache, myalgias, no side effects or other side effects". The variable was transformed into a binary outcome (Yes/No) for the quantitative analyses.

**Condom use.** Condom use was measured by asking participants the following question: "In which percentage of sexual encounters with clients and/or personal partners did you use condom yesterday?". Participants could drag the cursor left or right to choose the desired percentage (0% to 100%). The variable was transformed afterwards into a binary outcome to conduct quantitative analyses (Yes = use of condoms with all clients in all encounters that day, No = no use of condoms in all sexual encounters with clients that day).

**Number of clients.** Number of clients was measured by asking participants: "How many clients did you have yesterday?" to which participants replied by typing the desired number.

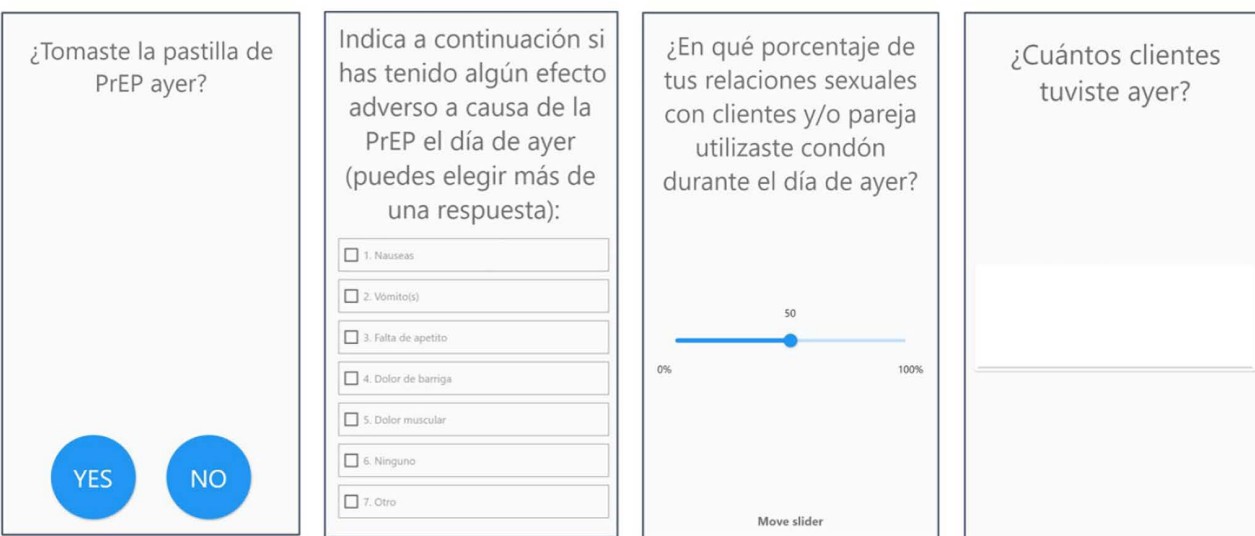

**Fig 1. Screenshots of the app-based daily questionnaire.** The first screen asked: "Did you take the PrEP pill yesterday?" with response options Yes or No. The second screen asked: "Please indicate below if you had any adverse effects yesterday due to PrEP (you can choose more than one option)," with selectable items including nausea/vomiting, lack of appetite, abdominal pain, headache, myalgias, no side effects, or other side effects. The third screen asked: "In which percentage of sexual encounters with clients and/or personal partners did you use condoms yesterday?" Participants selected a value from 0% to 100% using a sliding cursor. The fourth screen asked: "How many clients did you have yesterday?" and participants entered the number manually.

## Qualitative measures

The semi-structured interview guide (available at https://osf.io/6rdxh/) was developed based on existing literature and input from a multidisciplinary study team with expertise in HIV and sexual health among sex workers. The MAM served as a theoretical framework [40]. The guide explored motivations behind PrEP initiation and sustained adherence, the development of daily routines, and the impact of PrEP on work life and vice versa, with special attention to number of clients, condom use, drug use, stigma, and communication with clients and other sex workers. In addition to using open-ended questions (i.e., "Tell me how your experience taking PrEP daily has been"), the interviewer used probing questions informed by each participant's graphic summarizing their daily questionnaire results to help retrieve details not recalled spontaneously (i.e., "Let's talk about this day in which you reported not taking PrEP, can you tell me about the reasons or events that led you to not take PrEP that day?"). Particular attention was given to distinguishing whether episodes of nonadherence were intentional or unintentional.

## Data analysis

**Quantitative analysis.** Descriptive statistics and individual visualization tools, such as heatmaps and individual time-series plots, were used to characterize each outcome separately for each study participant. Descriptive statistics were also used at the group level (i.e., the participants were pooled together) along with generalized linear mixed effects models (GLMM). The latter were used to investigate the average effect of time on side effects, condom use, and the number of clients. After comparing several candidate models based on model fit, the final models included a fixed effect for study day (i.e., the progressive number of days in the study after PrEP initiation) and a random slope and intercept for each participant. Since the sample size was insufficient for formal comparison analysis between cisgender and transgender women; exploratory GLMMs were run with and without cisgender women. Results for the model without cisgender women were only reported when they diverged in direction or magnitude from the full model. Data were analyzed using the statistical software R version 4.2.0 (R Foundation for Statistical Computing). The dataset and code used for this study are available on the Open Science Framework platform (https://osf.io/6rdxh/).

**Qualitative analysis.** Interviews were transcribed and coded in Spanish and later translated into English to facilitate review and feedback by non-Spanish-speaking members of the research team.

A hybrid deductive-inductive thematic analysis was performed. LJVG created the coding framework in two phases: (1) main categories were established deductively from literature review and research aims, and (2) subcategories emerged inductively through preliminary analysis of interview transcripts. LJVG and CV then conducted the comprehensive qualitative analysis in MAXQDA [41], systematically applying the codebook, identifying patterns and relationships between codes, and interpreting themes across all interviews. The resulting kappa coefficient for each interview was ≥ 0.72 [42]. Data on the topics addressed in the current study was comprehensively summarized by LJVG and CV. At each step of the process, disagreements were discussed until a consensus was reached.

**Mixed methods analysis.** During the integration phase joint display tables were built to compare and contrast quantitative results with qualitative insights at the individual level. All tables can be found in the supplementary material file at https://osf.io/6rdxh/.

## Results

### Demographics

A total of 16 women sex workers were approached for recruitment for the study. Finally, 15 participants (12 transgender and 3 cisgender women) aged between 23–43 years old were enrolled between November 2022 and January 2023. All participants were immigrants from Latin America (Table 1).

## PLOS Global Public Health

**Table 1. Baseline characteristics of participants prior to starting PrEP.**

| ID | 1 | 2 | 3 | 4 | 5 | 6 | 7 | 8 |
|---|---|---|---|---|---|---|---|---|
| Gender | Trans. woman | Trans. woman | Trans. woman | Trans. woman | Trans. woman | Trans. woman | Trans. woman | Trans. woman |
| Age | 40-50 | 30-40 | 40-50 | 30-40 | 30-40 | 30-40 | 20-30 | 30-40 |
| Origin | Latin America | Latin America | Latin America | Latin America | Latin America | Latin America | Latin America | Latin America |
| Studies | Secondary school | Secondary school | Elementary school | Secondary school | Secondary school | University | Secondary school | Elementary school |
| Work place | Street, apartment | Street, apartment | Street, apartment | Street, apartment | Apartment, house | Apartment | Apartment, house | Street, apartment |
| Years in sex work industry | 25-30 | 15-20 | 25-30 | 1-5 | 1-5 | 1-5 | 1-5 | 1-5 |

| ID | 9 | 10 | 11 | 12 | 13 | 14 | 15 | |
|---|---|---|---|---|---|---|---|---|
| Gender | Trans. woman | Trans. woman | Trans. woman | Trans. woman | Cisgender woman | Cisgender woman | Cisgender woman | |
| Age | 30-40 | 30-40 | 30-40 | 40-50 | 20-30 | 20-30 | 30-40 | |
| Origin | Latin America | Latin America | Latin America | Latin America | Latin America | Latin America | Latin America | |
| Studies | Professional formation | Secondary school | Elementary school | Professional Formation | Elementary school | Elementary school | Secondary school | |
| Work place | Apartment | Street | Street, apartment | Street, Apartment | Apartment | Apartment | Apartment | |
| Years in sex work industry | 5-10 | <1 | 1-5 | <1 | <1 | <1 | 1-5 | |

We collected an average of 69% daily survey responses per participant (from 3% to 100%) (S1 Table). One participant (ID12) was lost to follow up and did not complete the exit interview, and another participant's interview (ID9) was not recorded. Additionally, one participant (ID15) only replied to 3% of the daily surveys but completed the exit interview. Because of this, we opted for removing ID 15 from quantitative analysis. After doing so the average daily responses per participant was 74%.

## PrEP initiation

Most participants heard about PrEP from other sex workers or through personnel from the non-profit organizations collaborating in the study. Others learned about PrEP online, including through dating apps such as Grindr. None reported receiving information about PrEP from healthcare workers, with one participant mentioning that her primary care physician confused PrEP with antiretroviral treatment (ART). When asked why they decided to start PrEP, most participants immediately mentioned condom rupture and/or condom removal by clients without consent. Using PrEP would also help avoid trips to the emergency room to take HIV tests, where they fear judgment of others in the waiting room. Less commonly, participants stated that using PrEP could help them retain clients who request condomless sex during periods of financial need, or allow them to enjoy sexual encounters with attractive clients without having fear or remorse about HIV risk.

The main and almost exclusive concern deterring participants from initiating PrEP was potential side effects. Additionally, some participants who had considered PrEP prior to this study did not start it due to limited PrEP knowledge among their regular care providers or long waiting times at the only designated PrEP center in the city. A summary of qualitative findings and selected participants' quotes can be found in Table 2.

**Table 2. Summary of qualitative findings and selected participants quotes related to PrEP initiation.**

| Broad themes | Main findings | Examples of comments |
|---|---|---|
| **Knowledge about PrEP** | Most participants heard about PrEP from other sex workers or through personnel from the non-profit organizations involved in the study. Other participants learned about PrEP online. | *"The woman from the association, she really helped me a lot. They gave me a lot of guidance and advice on how to take it, and they helped me so much, especially since I didn't have social security at the time. They really helped me get access to it."* - ID6<br>*"So through a friend I found out about the pill then, but she told me that it was for people who are like… promiscuous, then and people who are in prostitution. Then he gave me an idea and I started researching to look for it"* - ID4<br>*"I started following a page on Instagram that talked about PrEP, and I would see comments about it. I also downloaded this app called Grindr, which is for dating. So, while looking for casual encounters, I noticed that many profiles mentioned "I take PrEP." One day, I reached out to someone, not to meet up, but to ask about the pill, and they explained it to me."* - ID4 |
| **Reasons to start PrEP** | Ninety percent of respondents gave condom rupture and/or condom removal by clients without consent as their first intuitive response to why they started taking PrEP. PrEP was regarded as an extra layer of protection to regain control of something that used to be 'out of their hands'. Less commonly participants stated economic gains and personal pleasure as motivators to start PrEP. The main and almost exclusive concern deterring participants from initiating PrEP was potential side effects. | *"The biggest fear is that a condom might break, or that since sometimes with clients you take drugs, or you're not exempt from being drugged or drinking alcohol, that in one of those moments of losing consciousness, something happens, and they penetrate you without a condom or something like that. That's the biggest fear."* - ID6<br>*"Or sometimes the guys, you don't realize, and they say, 'get on all fours.' I hate getting on all fours because there are guys who are so sneaky that by the time you've turned around, they've already taken off the condom. So whenever I get on all fours, I always touch them to make sure they have the condom on, or sometimes they can break."* - ID4<br>*"No, me, because I liked to have unprotected sex and when I did, I did feel at that risk, in that fear of agony that I couldn't even sleep because I thought and had that doubt that I could have HIV and since I'm taking it (PrEP), for me it's the best"* - ID2<br>*"From time to time one makes mistakes... there are clients who offer you double what they are paying you, so that you do not use the condom..."* - ID3 |
| **Barriers to PrEP initiation** | The main and almost exclusive concern deterring participants from initiating PrEP was potential side effects. Lack of PrEP knowledge by their regular care providers, or long waiting times at the only designated PrEP center in the city were barriers to start PrEP prior to the current study. | *"… I'm going to be honest, at first I thought about it because of the changes I was going to see, you understand? because using something unknown. It's like you're a little afraid of it."* - ID4<br>*"At one point, I thought about PrEP, but then I decided not to because I had heard that hospital appointments were scheduled for 3, 5, or even 6 months later... So, I never made up my mind"* - ID1 |

## PrEP adherence and side effects

Individually reported PrEP adherence ranged from 37% to 100% (mean 68% among transgender women and 89% among cisgender women; S1 Table). Missed doses appeared randomly distributed throughout the study (Fig 2) with no significant effect of time on adherence (GLMM estimate -0.06 95% CI -.04 to.03). Side effects prevalence ranged from 0% to 66% (mean 26.4% among transgender women and 32.9% among cisgender women; S1 Table), and decreased modestly over time (GLMM estimate -.06, 95% CI: -.13 to.00; Fig 2). Among all days with reported symptoms, nausea and vomiting accounted for 30%, low appetite 6.6%, headache 14.1%, abdominal pain 11.5%, myalgias 10.6%, and other symptoms 44.6%.

Qualitative data helped explain participants' strategies to remain adherent and circumstances driving short-term variations. Adherence was supported by the creation of daily routines, like choosing a top-of-mind PrEP storage location, linking dosing to meals, using alarms, or taking it alongside other daily medications. In contrast, participants with irregular adherence described side effects and work-related factors such as long night shifts, substance use, and travel as barriers to maintaining routines, mirroring the scattered missed doses observed in the time-series data. Intentional non-adherence also emerged, often motivated by concerns about mixing PrEP with alcohol, recreational drugs, or other medications; by

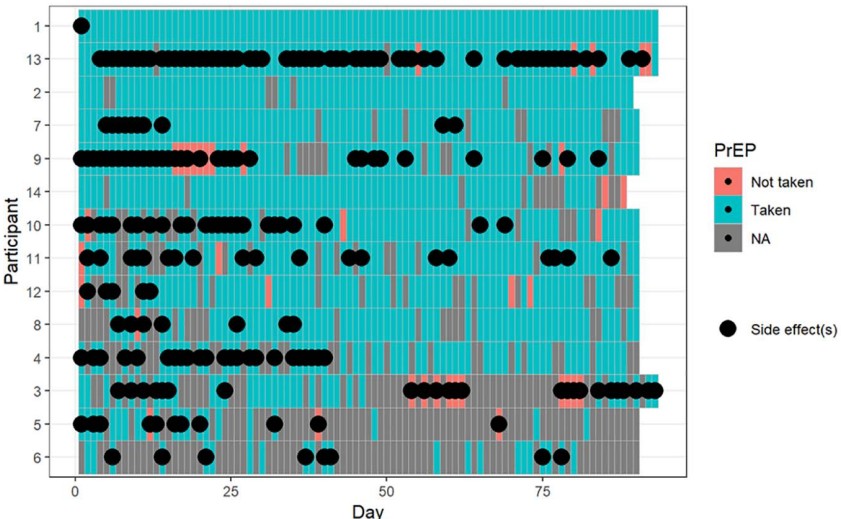

**Fig 2. Heatmap representation of PrEP adherence and side effects during the study period.** The Y axis represents individual participants, and the X axis represents days since study initiation. Color coding: blue indicates days when PrEP was taken, orange indicates days when PrEP was not taken, and gray denotes missing data. Dots mark days on which a side effect was reported.

wanting to avoid PrEP-related side effects that would otherwise difficult their work (e.g., diarrhea); or by avoiding doses taken "too close together". Recreational drug use was more commonly associated with intentional nonadherence than with forgetfulness. Longer interruptions (more than three days) were related to illness, and for one patient a scheduled surgical procedure. Participants did not engage in sex work during these long pauses and respected a lead-in time after resuming PrEP. Participants' expressed frustration with the restricted access to PrEP, as refills were only available at the hospital's pharmacy by prior appointment, although this did not lead to missed doses. When integrating qualitative and quantitative findings, participants' reported PrEP intake was consistent with taking four or more doses per week during periods of sex-work activity. Only one participant stopped taking PrEP entirely, coinciding with her exit from sex work.

Participants used several strategies to manage side effects and sustain adherence, including adjusting the timing of doses (e.g., taking PrEP at night or with meals), relying on the expectation that side effects would diminish overtime, and using over-the-counter proton pump inhibitors. In addition, most participants felt a sense of personal responsibility and pride in taking care of themselves as both a motivator and a benefit of PrEP intake. A summary of qualitative findings and selected participants' quotes can be found in Table 3.

A joint display of results integrating quantitative time-series data and qualitative insights for each participant can be found in S2 Table

**PrEP use and sex-work related factors: Condom use, number of clients, and PrEP-related communication**

Across the study period, condom use varied widely between participants, ranging from 9.3% to 100% (mean 57% among transgender women and 100% among cisgender women; S1 Table). At the group level, condom use did not change significantly over time after PrEP initiation (GLMM effect estimate 0.01, 95% CI: -0.01 to 0.03; Fig 3).

Qualitative data provided complementary information about sexual practices after PrEP initiation, with a decrease in the frequency of condom use with clients commonly reported. Participants described several reasons for this decrease, primarily economic gains, followed by pleasure, and for one participant, a desire to test PrEP's effectiveness. Participants described being selective about condomless sex, which was reserved for "loyal clients", clients who paid "very well" for

**Table 3. Summary of qualitative findings and selected participants quotes related to PrEP adherence and side effects.**

| Broad themes | Main findings | Examples of comments |
|---|---|---|
| Facilitators of PrEP adherence | Participants' main strategy to remain adherent to PrEP was the creation of new routines. Choosing a top-of-mind PrEP storage location was a key strategy to create a daily habit of taking the pill. Taking PrEP around mealtimes was another strategy for some participants to avoid side effects and remember taking it; however, for others it became difficult since their line of work created irregular mealtimes, leading to skipped doses. Personal alarms and notifications from the daily surveys had mixed efficiency. Some participants found easier to remain adherent to PrEP when they were already taking other medications. Finally, participants motivation to remain healthy and deep sense of responsibility towards their health facilitated adherence. | "What I did was keep them on the bedside table with the boxes, so when I saw them, I knew I had to take them." - ID4<br>"Not always, like I said before, I would take them starting at 10:00 in the morning, after breakfast, but sometimes I would forget and take them after lunch instead." - ID8<br>"At first, I did have reminders when we were doing the survey. Mostly to remind myself to take the pill around the same time as the questionnaire. Then it became a habit."- ID7<br>"Well, I take it with vitamins, really well. So, I take it with those, and I feel... I have complete peace." - ID3<br>"I have it as a priority. It's just that I'm so happy with that medication that I don't put it (the alarm)... I assumed it as part of me, as if it is a medicine that will favor my health, my well-being, my safety, my work, me, my emotions, my day to day. And I have more than assumed it. It's already part of my life, that I know that every day on a schedule I take my medication." - ID2<br>"…because in the end what motivates you is that you are calm, that in spite of all you go to work calm because you don't know the desperation that I had before the pill."- ID4<br>"I set my alarm, my alarm goes off is: "for my pill, for my pill, for my pill."- ID8<br>"I really want to protect myself because I have my family waiting for me. I know there's someone waiting for me in my country."- ID10<br>"The truth is that since I started using PrEP and I became much more aware, that is… much more aware of the risks that I took before and I don't know, it feels as if it disciplined me more because every day I have to be taking the pill and I have to take it seriously too and I work with people and many men are married. So you know what? That's how it turned out... it disciplined me much more."- ID11 |
| Barriers to PrEP adherence | Side effects were the main barrier to PrEP adherence. Participants interrupted PrEP when they're feeling very unwell out of fear that it could exacerbate their symptoms, or due to the presence of gastrointestinal symptoms that difficulted optimal adherence. Forgetfulness around taking PrEP was highly correlated to changing work schedule, leaving their residency at sporadic times, travel, and sleeping through their pill time. The use of alcohol or drugs, also impacted PrEP adherence. Sometimes it would lead to forgetfulness, and other times participants would consciously opt to skip their PrEP dose to avoid mixing it. When participants modified the time they took PrEP from day to day due to forgetfulness or side effects, this led to skipped doses in order to avoid "taking them too close in time". Finally, restricted access to PrEP only at authorized healthcare centers required participants to plan well ahead of time when to obtain their refills, which was an added layer of difficulty. | "At first… With the changes and all that… you tell yourself what's going to happen in your body or how you're going to react, so that generates anxiety"- ID4<br>"The negative part was the symptoms at the beginning. The first few days. Dizziness and headaches. Very very strong headaches… Because of it, and since I was not working, I would take the pill every other day" - ID6<br>"Because of the asthma, I was really unwell, and I had to stop it because I couldn't take the pill or anything. I had a very bad cold, I was extremely congested. And I said, look, I can't because it (the PrEP) irritates me and I just can't. I would vomit, and I was feeling really bad and couldn't manage. During that time is when I missed taking the pill." - ID5<br>"…since they prescribed me a very strong antibiotic and on top of that I was taking the pill (PrEP), then (I knew) that was going to do bad to me... And it hit me, because I got gastroenteritis with all that, because as I was taking that very strong medication, my defenses were lowered and I fell, I fell ill immediately with the flu and I had about 15 days that I couldn't eat... Then I had to stop the pill and now I took it again. After I was cured of everything, I took it again, the seven days to put up with the (murmur) until I..." - ID4<br>"It was work-related stuff, darling, yes, I'm working many hours and I'm quite busy, you know? Without eating, without... that's why I didn't take it so I wouldn't feel bad." - ID11<br>"When one is working. Well, when one has diarrhea, it's a little uncomfortable to have to work like that." - ID10<br>"No, there are times when, like I was telling you, the schedule (work related schedule)... I've spent the whole time sleeping."- ID10<br>"And yes, other times it was because I went out and didn't bring the pill with me, so it was more because of that. Between the time I remembered to take it and when I got home or wherever I was, I would take it again. The next day there was very little time between doses, so I didn't want to take them so close together in terms of hours." - ID13<br>"And I had them in his suitcase. And I was up in the cabin. I was in the cockpit, and I was up on the plane. And it happened to me twice one weekend that I left and… And I didn't take them... …It took me three days because it was Friday, Saturday and Sunday, until Monday when I got home I didn't take them" - ID6<br>"The times I've been really high, there have been times when I didn't take PrEP precisely because I was on a lot of drugs and was a bit afraid of... you know? Mixing them and having some kind of effect. So no, it's like, with all the effects and everything, I prefer not to take it because if taking it is going to make me feel worse, I don't want to harm my body even more, that's why." - ID1<br>"Suddenly you can spend so many hours of the night and in the early morning working or under the effects of drugs, that the next day you forget to take PREP because you missed lunch time and you only get up to do something and sleep. That's why" - ID7<br>"Because we trans girls and workers travel all over Spain. So with a card you can claim it in any city. That would be very, very, very good." - ID6<br>"…that if they could give it in... What it's called, in any pharmacy, I mean, pharmacies, local pharmacies, instead of having to come to the hospital to get it and all that. Do you understand me? That influences a little sometimes too" - ID1 |

*(Continued)*

Table 3. (Continued)

| Broad themes | Main findings | Examples of comments |
|---|---|---|
| **Coping mechanisms** | Participants relied on the knowledge that side effects from PrEP tend to disappear over the course of few weeks, and a sense of personal responsibility and motivation. Some participants discussed this with close friends, coworkers or medical personnel. Other times coping mechanisms evolved around choosing the time of day to take the PrEP (with meals or before bedtime) to minimize the impact of the side effects. Some participants relied on the consumption of proton pump inhibitors and acetaminophen to combat the side effects. | *"And the girl from the association also told me "(name) no, that's is, that's one or two weeks, but your body is going to assimilate it, nothing is going to happen" and from then on I didn't feel it. When I stop taking it and I take it again, I don't feel anything either"- ID6*<br>*"… I told her that I was taking this thing (PrEP) and she said to me "What effect did it give you? It gave this to me" and I told her "Me too", that is, it is just a conversation, but then I was relieved because I knew that it had not only happened to me" - ID4*<br>*"It's just that I drank it first in the day, after lunch. Then I felt groggy, and I got nauseous. Then I said I'm going to take it before eating, let's see if suddenly it's because I have a full stomach that I don't like it …. But not…with an empty stomach it burns me, it burns with fire, so no. So I decided to take it at night and I got dizzy, I got my headache, and I went to bed and that's it."- ID6*<br>*"(I recommend) that they accompany it (PrEP) with a stomach protector, because the truth is that the pill falls very heavy at first".- ID6*<br>*"If I knew that it was going to have possible effects and such, I avoided consuming any type of food that would make the situation worse for me. If I had diarrhea I tried not to drink whole milk and things like that" - ID7*<br>*"To take it like at dawn, when I was already coming home from work, so I if I had anything (side effects) then I was already inside at home. You know?" - ID10* |
| **Stopping PrEP** | Only one participant decided to stop taking PrEP, and it was due to her quitting sex work. | *"Because I was going to leave the job I was in. I had already withdrawn more from that and now I am no longer in that world. I've already retired." - ID14* |

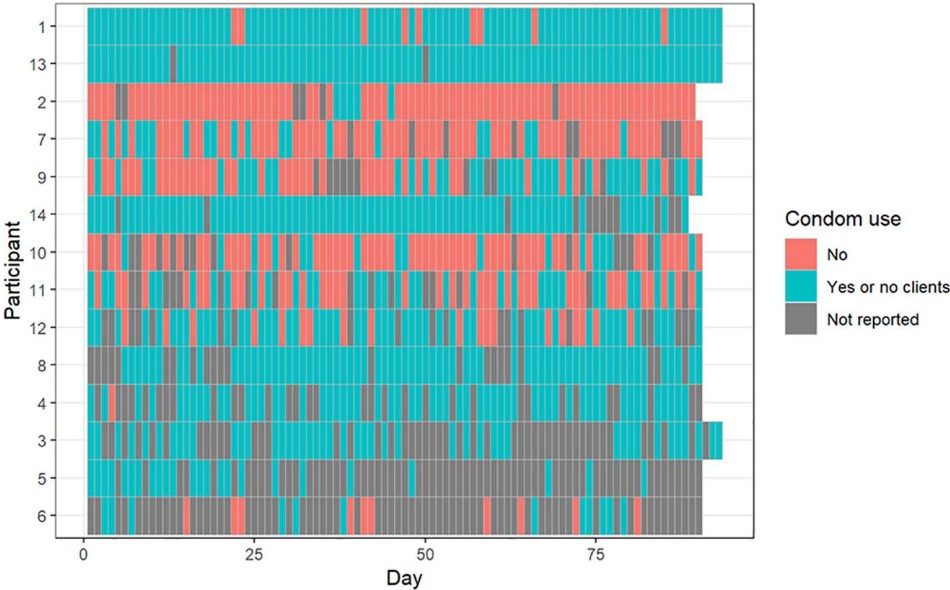

**Fig 3. Heatmap representation of condom use with clients during the study period.** The Y axis represents individual participants, the X axis represents days since study initiation. Color coding: blue indicates days when a condom was used in all sexual encounters or when no clients were reported; orange indicates days when condom use occurred in less than 100% of sexual encounters; and gray denotes missing responses.

their services, and occasionally clients perceived as particularly attractive. For some participants, reduced condom use involved condomless penetration, whereas for others it referred only to oral sex and/or genital rubbing. Fear of non-HIV STIs remained a major reason to continue using condoms in most sexual encounters. Participants who acknowledged a decrease in condom use after initiating PrEP also reported high PrEP adherence in their daily questionnaires (>80%). A summary of qualitative findings and selected participants' quotes regarding the impact of PrEP on condom use can be

found in Table 4. A joint presentation of quantitative and quantitative findings regarding changes in condom use after PrEP initiation for each participant can be found in S3 Table.

The number of clients reported through the daily questionnaires remained stable throughout the study (GLMM effect estimate -0.00, 95% CI: -0.01 to 0.00; Fig 4), with transgender women reporting clients 60.7% of days (mean 2.6 clients per day) and cisgender women 39.9% of days (mean 2.5 clients per day) (S1 Table).

Most participants reported that variations in the number of clients after starting PrEP were unrelated to PrEP use and were instead attributed to other external factors such as festivities, economic conditions, changes in sex-work policies that occurred at the time of the study, and travel. Nevertheless, some participants noted that relying on PrEP to engage in condomless oral or penetrative sex increased the number of visits from certain clients. A summary of qualitative findings

**Table 4. Summary of qualitative findings and participants quotes regarding the impact of PrEP on work life.**

| Broad themes | Main findings | Examples of comments |
|---|---|---|
| Condom use | A decrease in the frequency of condom use with clients was common among participants after initiating PrEP for economic gains, personal gains, and for one participant, to test if PrEP works. Condomless sex usually involved some of the "loyal clients", clients who paid "very well" for their services, and occasionally clients considered very attractive by study participants. While some participants agreed to penetration without condom, to others, the decrease in condom use referred only to oral sex and/or rubbing of the genitals. Fear of STIs other than HIV was a major reason to continue using condoms in most sexual encounters. | "I think (I did it) to prove that the drug works" - ID1<br>"Because I know that if I don't use a condom, even if I don´t get HIV, other types of diseases can get on me. So generally, most of the time, ninety-odd percent, I use condoms" - ID1<br>"Not in the past and now, now that I can do it and I have even had clients who have wanted to pay me and no, and I do not give in. I prefer to put protection and continue protecting myself. I only catalog a benefit that can bring me for the economic part and for the other sexual part that I am going to have a great time. I catalog the type of boy that I like"- ID2<br>"Let's see, I haven't used a condom because they are guys I know, people I see regularly once a week" - ID2<br>"From time to time maybe yes (I have condomless encounters) because of the issue of necessity, it always happens that you agree to have unprotected sex for an economic amount, you know?" - ID7<br>"In other words, there are customers who like to touch it, play... But most of the people with whom I do this are with clients that I already have permanently"– ID4 |
| Clients | Overall, participants reported that changes in their number of clients after starting PrEP were unrelated to PrEP use itself and were instead driven by other external factors. Nevertheless, some participants noted that relying on PrEP to engage in condomless oral or penetrative sex increased the number of visits from certain clients. | "Because there are good days and bad days. For me, the best days are the first days of the month. Because that's when people get paid."- ID4<br>"The number of clients has nothing to do with PrEP." - ID8<br>"Well, yes, they come more often. Yes… it gives me more. And they, because in the past I didn't want to do it and they, having the knowledge that I can now do it without protection, come more frequently, they call me more, yes." - ID2<br>"It has favored in the factor that I tell you that there are many who like oral sex without a condom and since I did not do it they did not come. Now that I allow myself, more are coming"- ID6 |

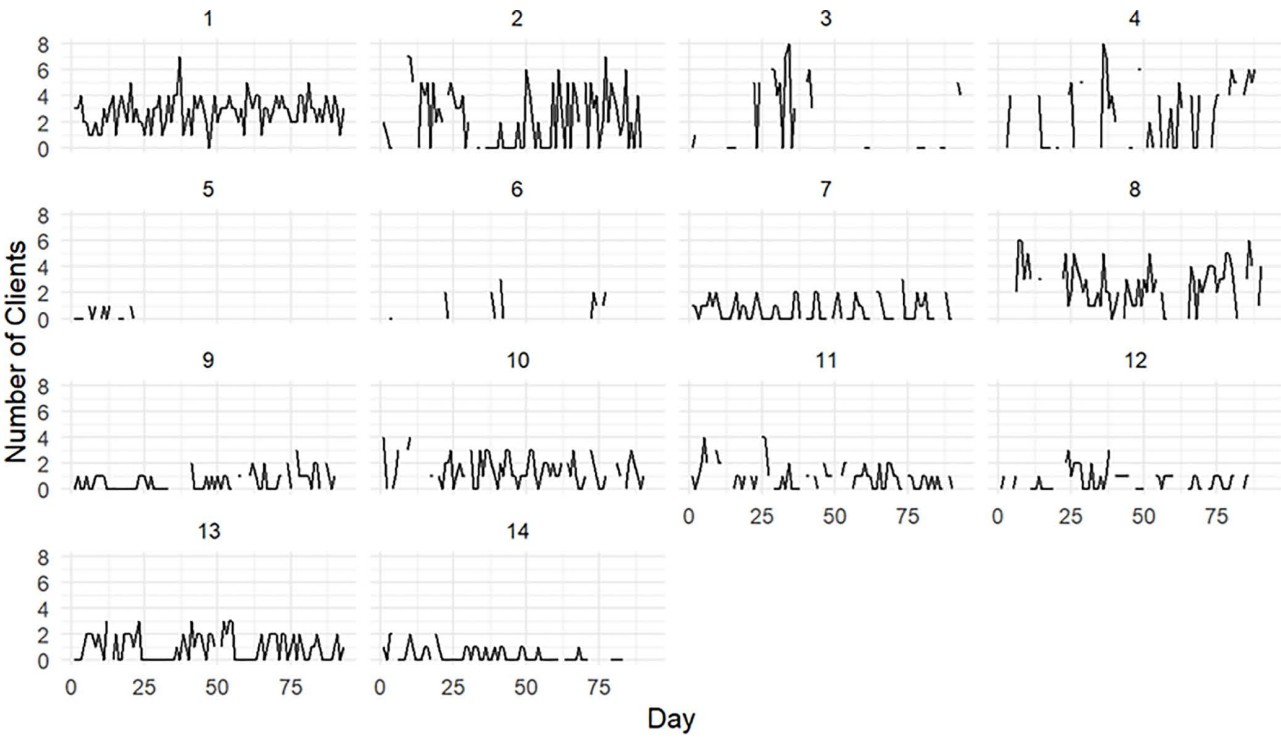

**Fig 4. Time series representing the number of clients for each individual participant during the study period.** Each panel represents an individual participant, labeled from 1 to 14. The X axis indicates days since study initiation, and the Y axis shows the number of clients reported per day (range: 0-8).

and selected participants' quotes regarding the impact of PrEP on the number of clients can be found in Table 4. A joint presentation of quantitative and quantitative findings regarding changes in number of clients after PrEP initiation for each participant can be found in S4 Table.

Consistent with these findings, participants disclosed PrEP use to only a small number of clients: long standing clients they knew and trust, as well as "higher-end" clients specially concerned about safe sex. Notably, the cisgender women in the study chose not to disclose their PrEP use to any clients. For those who did disclose, sharing PrEP use with selected clients was described as a way to provide reassurance and maintain loyalty. Participants feared that open disclosure could encourage clients to request condomless sex, which they wished to avoid. Participants also perceived that many clients sought condomless sex regardless of STI risk and were unlikely to value information about PrEP, especially when interactions were quick and succinct. Moreover, there was a concern about clients confusing PrEP with ART, which would create additional problems.

With regards to PrEP communication with coworkers, trust within the sex-work community was described as limited. In communities with high PrEP awareness, participants feared judgment from coworkers who might assume they engaged in "reckless sex", whereas in communities with low awareness, they feared being judged due to confusion between PrEP and ART. One participant noted that PrEP-related stigma among sex workers often stemmed from perceptions of the so called "PrEP girls", perceived as offering condomless sex at lower prices, something that was felt to negatively affect the sex-work market. However, PrEP also created a sense of pride for a minority of outspoken participants, who wanted to become advocates for the pill in the sex work community. A summary of qualitative findings and selected participants' quotes regarding PrEP communication with clients and coworkers can be found in Table 5.

**PLOS Global Public Health**

Table 5. Summary of qualitative findings and participants quotes regarding PrEP communication with clients and coworkers.

| Broad themes | Main findings | Examples of comments |
|---|---|---|
| Communication with coworkers | Participants were selective sharing their PrEP experience with other sex workers due to stigma. In settings with high PrEP awareness, they worried others would assume they engaged in "reckless sex" while in settings with low awareness, they feared being judged because PrEP might be mistaken for ART. PrEP also created a sense of pride for a minority of outspoken participants, who wanted to become advocates for the pill in the sex work community. PrEP-related stigma amongst sex workers often came from the so called "PrEP girls" perceived as having sex without a condom for a lesser price, depreciating prices and taking away clients. | "most of them have told me 'that's for bareback fucking', but well, 'You don't do it?', I say 'No, no, I don't. I do it as a precaution'… I tell them 'you that presumably take care of yourself so much. You don't take any pills, your rubber breaks. What are you going to do? Going to the emergency room or being at home all day the day after thinking what's going to happen?'… And all that. So, I tell them 'to avoid those problems, I tell myself, I take the pill" - ID 4<br><br>"I'm a girl who, at any conference I attend - in fact, I was in an association, where they were discussing the topic - I stood up and said, 'Look, I'm a girl who is on PrEP.' Like I said, I expose my situation, I expose myself and my reality because I'm sure I don't have HIV, and this is a medication that prevents HIV." - ID 2<br><br>"So there are many who don't, they don't take it (PrEP), because 'ahh if you take it, you like to fuck condomless', but it's not that, it's like a... I think it's like a protection, you have to protect yourself." - ID 10<br><br>"There are people who don't have that information and think that you are the one who has HIV." - ID 2<br><br>"So, like I said, there are PrEP girls, that's what we call them, because they're on PrEP, and they agree to everything and they're young"- ID 3 |
| Communication with clients | Participants reported taking PrEP to protect themselves from clients who look for consented or unconsented condomless sex. They also felt that most clients would not appreciate information about PrEP, as clients were perceived as largely unconcerned about STI transmission risks. Consequently, most participants chose not to disclose their PrEP use to clients, or did so only with a select few to secure or encourage repeat visits, typically long-standing clients they trusted or "higher-end" clients who were particularly concerned about safe sex. This was viewed as an indirect benefit of PrEP use rather than a direct motivator for initiating or maintaining PrEP. | "If I tell them yes to penetration without a condom, my house will fill up. Everyone wants that... But it is very dangerous. I don't take risks because there are other diseases" - ID 6<br><br>"When the client comes to do it with you without protection, he does not have an opinion. The client, what he wants is to do it without protection, the client is not caring, the client does not want protection. Already that type of customer, why are you going to give them additional information that they are not looking for? Because time is short. I give additional information when I already know the person and we have that empathy, but a regular day-to-day client who is looking for unprotected sex. Imagine that I am a customer and you are a girl If I tell you, 'I want to do it without protection', why are you going to cut me off? 'Look, I'm going to do it to you without protection...' it doesn't fit, because what I'm looking for is to do it without protection. Let's get down to business, that's it" – ID 2<br><br>"What I'm trying to say is that when you tell them you're on that treatment, they suddenly assume you have some kind of illness or something like that."- ID 8<br><br>"… there are people who suddenly say, 'Look, I'm married, I'd like you to tell me a bit if you take care of yourself, if you're on any treatment, or if you have any disease.' I tell them, 'Look, I don't have any disease, here are my test results'…" - ID 5<br><br>"I explain it to them (saying that I take PrEP) because these are guys who have been with me for years, and I want to be honest with them so they don't leave with any doubts, because you could have that doubt of 'What if she has HIV?' And what I want is for them to leave feeling at ease." - ID 2 |

Study results mapped to the Medication Adherence Model can be found in Fig 5.

## Discussion

We enrolled 12 transgender and 3 cisgender women sex workers, all immigrants from Latin America, in an intensive longitudinal mixed methods study to capture their experience during the first three months of PrEP use. Average reported PrEP adherence was above 70%. Adherence was supported by personal motivation, daily routines, and the use of alarms. Missed doses were randomly distributed and reflected both unintentional non adherence (side effects, work-related disruptions) and intentional non-adherence (avoiding side effects that could affect work performance). Most participants relaxed their use of condoms with few selected clients based on "loyalty", high economic reward or personal attraction,

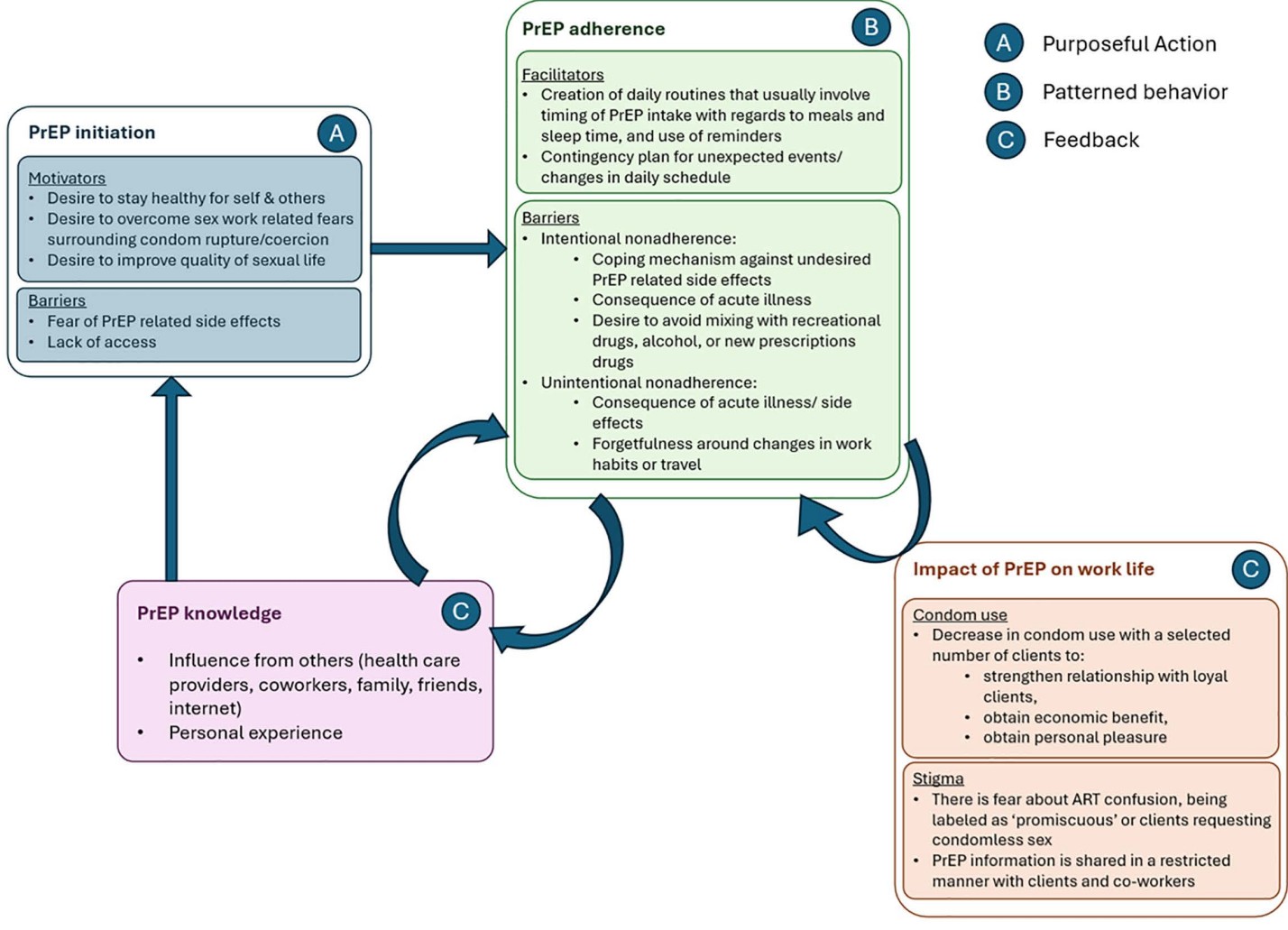

**Fig 5. Study results mapped to the Medication Adherence Model.** This figure organizes participant-reported experiences into three domains of the Medication Adherence Model: Purposeful Action **(A)**, Patterned Behavior **(B)**, and Feedback **(C)**.

although this did not appear to affect the overall number of clients. PrEP and HIV related stigma were common among the sex worker community.

In alignment with prior research [12,43,44] participants heard about PrEP from other sex workers, online apps, or NGOs. Lack of knowledge and stigma probably explain why health care professionals are not a primary source of PrEP knowledge for women in Europe [45]. The main reason participants chose to start PrEP was to gain control and be reassured that they are protected from the 'main', incurable disease, HIV. This is concordant with what was already described in a cross-sectional study among female sex workers in Madrid, Spain [12], and other women sex workers around the world [22,46].

Current literature shows that PrEP adherence among women sex workers is usually low, with high rates of early discontinuation [17,33,34,47]. To date, only two studies have examined longitudinal PrEP trajectories among cisgender and transgender women sex workers. One longitudinal study among female sex workers in South Africa described frequent cycling on and off PrEP amongst their participants, although it did not clarify whether interruptions reflected changes in

HIV risk [48]. The second, in the United States, described several distinct patterns in how transgender women stopped and restarted PrEP, One of these patterns was "PrEP cycling", which tended to follow changes in HIV risk linked to shifts in socio-behavioral circumstances, such as moving out of and back into sex work. Importantly, this trajectory was not associated with HIV seroconversion, suggesting that it may represent a prevention-effective adherence pattern [49].

Our current study adds insight into longitudinal patterns of daily oral PrEP use among women sex workers. Only one participant discontinued PrEP entirely, coinciding with her exit from sex work. Despite variable levels of adherence, participants' reported PrEP intake appeared sufficient to preserve efficacy (four or more doses per week) [10,13]. Participants appeared to practice prevention-effective adherence: missed doses were randomly distributed, and when participants missed PrEP for several consecutive days, they did not engage in sex work and remained cautious after restarting the medication. These findings highlight that women sex workers are capable of making informed decisions about PrEP use, modulating adherence according to perceived HIV risk, cycling off during lower risk periods, and respecting lead-in times before resuming high risk activity. These findings also suggest that early adherence challenges are driven by day-to-day circumstances rather than progressive disengagement. These results should be interpreted within the supportive context of an observational study with adherence monitoring. Continued collection of this type of information will remain important even as injectable PrEP products are implemented, since adherence gaps between injections will be threats to efficiency [49]. Such data can inform appropriate counseling and support, and may also contribute to evaluating the potential role of event-driven PrEP use among cisgender and transgender women (not yet approved by regulatory agencies).

Nevertheless, we identified some opportunities to boost adherence among study participants. As seen in other studies [22,46], the main and almost exclusive concern deterring study participants from initiating PrEP was potential side effects, which were also a primary barrier to PrEP adherence. Based on our results, we hypothesize that women sex workers might be more likely than other PrEP users to skip PrEP doses due to side effects, particularly when symptoms interfere with their ability to work. Most of the copying mechanisms described in our study are described in the literature for men and transgender women who have sex with men: taking PrEP with meals, adjusting time doses, and seeking support from peers [50]. Additionally, study participants relied on the use of over-the-counter medications: acetaminophen and proton pump inhibitors. Providers are encouraged to discuss the potential use of these medications to help with early side effects at the time of PrEP initiation.

Moreover, our study provides insight on the different ways in which sex work might complicate consistent PrEP adherence [16]. On one hand, changes in daily routines between working and not working days lead to forgetfulness or skipped PrEP doses to avoid taking them too close in time. The future integration of long-acting formulations into PrEP clinics, recently approved by the European Medicines Agency in 2025, is a promising strategy to reduce nonadherence related to forgetfulness and routine disruptions [51]. In addition, the use of alcohol and drugs with clients impacted PrEP intake, sometimes because it made them more forgetful, but more often because participants intentionally skip PrEP to avoid unintended side effects of mixing substances. Previous research has associated drug use and binge drinking with poor adherence to PrEP or ART due to increased forgetfulness [52,53]. However, this might be the first study to confirm intentionally missed PrEP doses around the time of alcohol or drug use. This finding highlights the importance of harm reduction strategies within PrEP programs attending sex workers, but also that more investigation is needed to understand potential interactions between PrEP and drugs so that PrEP users can be adequately informed. Along the same lines, participants had concerns about interactions between PrEP and new prescription medications (e.g., antibiotics). The use of a user-friendly online application to check on drug-drug interactions could help answer these questions that arise between clinic appointments.

Finally, work related travel, especially prevalent among transgender women sex workers [8], lead to missed PrEP doses. None of the study participants missed more than a few days of PrEP. However, if this had happened during a work travel of long duration, strict regulations regarding PrEP dispensation at hospitals would have prevented sex workers from refilling PrEP at their destination [3], a particularly concerning issue given the elevated HIV risk during travel. Facilitating

PrEP dispensation by regular pharmacies, sexual health clinics, or NGOs, and developing formal communication channels between PrEP providers from different areas in Spain could help in these situations.

The report of the three cisgender women in our study aligned with prior studies that show no evidence of decreased condom use after PrEP initiation among women sex workers [54,55], A larger sample might have disclosed different behavioral patterns, as was the case with transgender women sex workers: some reported compensating behaviors, while others did not. The qualitative interviews provide more granular information about fluctuations in condom-use frequency and number of clients, which, despite not being large enough to have statistical significance, had a notable impact on participants' work life. Indeed, some participants experienced secondary gains by agreeing to condomless sex in situations of economic necessity or high economic reward, and less commonly talked about in the literature, by obtaining personal pleasure. These secondary gains appeared to support PrEP adherence.

Additionally, participants reported being affected by sexual practices of other coworkers using PrEP, who were believed to lower market prices by engaging in condomless sex with a large number of clients for a low fee. Therefore, it is possible that widespread use of PrEP could indirectly increase rates of unprotected sex among sex workers seeking to remain competitive in the market. Modelling studies of market forces after the introduction of PrEP have hypothesized a substantial mitigation of PrEP impact due to condom substitution [32], However, data about changes in sexual behaviors among women sex workers in Spain remain too limited to make any predictions [16,56].

Once again, the behaviors of the cisgender participants aligned with the limited existing knowledge about PrEP communication with clients: avoidance due to HIV stigma and fear of unsolicited condomless sex [57]. Little is known however about disclosure practices among transgender women sex workers. Our study informs that while similar dynamics appear to apply with most of their clients, some transgender women sex workers acknowledged that sharing information about their PrEP use helped them obtain more visits and foster a trusting relationship with a selected minority of clients.

Likewise, PrEP communication with other sex workers was affected by anticipated PrEP and HIV stigma. Some participants participated in these stigmatizing views, differentiating their behavior to those of "other" PrEP users. PrEP communication was often limited to a close circle of friends (often sex workers) who offered support by sharing knowledge, experiences, and reminders to take the pill. This highlights the crucial role of peer support to disseminate awareness and boost PrEP adherence. However, this information was not always true or accurate, supporting the need for official channels to verify information. Importantly, two participants became open "advocates" for PrEP in their community. PrEP programs should study if enrolling this profile of PrEP users for peer navigation, as recommended by the International Antiviral Society-USA [58], could help overcome barriers to PrEP engagement and adherence among sex workers.

Finally, prior studies have emphasized the positive mental health effect of PrEP among non-sex workers, linking PrEP use to a decrease in anxiety and depression [59,60]. These benefits in the sex work community, which is highly affected by mental illness [12,61], are very relevant. In our study PrEP fostered a feeling of responsibility among participants, who became more attentive to their own health and felt a duty to achieve good adherence.

### Strengths, limitations and perspectives

Firstly, the small number of participants limits the generalizability of our findings and precludes formal comparison between cisgender and transgender women sex workers. Nevertheless, each participant contributed a substantial amount of data (a total of 1,266 daily survey responses were collected), which strengthens the robustness of the findings. Enrollment figures were also consistent with clinic attendance patterns between 2022 and 2023 (approximately 4.2 women sex workers per month, cisgender-to-transgender ratio 0.2:1) [62], suggesting representativeness. Moreover, attendance at this clinic was also higher than in other national cohorts [16,63], likely reflecting the strength of the referral network. Because of this we hypothesize that sex workers attending the clinic likely felt comfortable disclosing their status, although we cannot assume all did so, as subsidized PrEP could also be accessed through other risk factors.

Secondly, the disproportionate ratio of cisgender to transgender women sex workers enrolled may reflect a combination of structural and contextual factors. Reports from European sex worker organizations describe limited PrEP awareness among cisgender women, lower perceived applicability of PrEP by healthcare providers, and continued reliance on condom-based prevention messaging [64]. In Madrid, prior surveys similarly found that a sample of predominantly street-based cisgender women sex workers had lower PrEP awareness and preferred mobile services [12], whereas transgender women were more familiar with PrEP and favored hospital-based clinics [13]. These differences in awareness, service preferences, and perceived HIV risk may have contributed to the higher representation of transgender women in our study. Nevertheless, the purpose of this innovative mixed methods study was not to stablish a comparison between groups, but to highlight each individual's PrEP journey, an information that complements that obtained in studies using traditional, group-level only results, and brings us closer to clinical practice.

Thirdly, the study sample consisted exclusively of women from Latin America, an under-researched group in terms of PrEP perceptions. Migrant women, particularly those with irregular status, may rely on informal information channels due to mistrust of institutions or limited knowledge of official channels [65,66]. Experiences of stigma and violence in their home contexts may further heighten vulnerability and shape engagement with prevention services in Spain [67]. Spanish-born women, who represent only a minority of sex workers reached through NGOs [68], may have acquired PrEP knowledge through different channels and displayed distinct adherence patterns.

Fourthly, the lack of data prior to PrEP intake, e.g., via a pre- post- study design, hinders the evaluation of its impact on the study outcomes. However, changes were expected to occur gradually after initiation of PrEP, which could be captured by analyzing the effect of time. Additionally, even though efforts were made in the study design to decrease recall bias, this is still a possibility; diaries or more frequent interviews may help decrease this in future studies. Social desirability bias is also possible, though personnel were trained to interact non-judgmentally, and serum tenofovir level tests to confirm participants adherence was not performed.

Finally, the qualitative results may have been influenced by researchers' backgrounds. Interviews were conducted by a medical student, a pharmacist, and an infectious diseases physician who had received training in interview techniques. The physician was also responsible for enrolling participants, which may have affected how some women responded; however, her clinical experience and long-standing work with this population enabled more nuanced, context-sensitive questioning. The decision to include the physician as an interviewer was made to address scheduling difficulties between some participants and the other two interviewers. Qualitative analysis was carried out jointly by a specialist in qualitative research methods and the same infectious diseases physician, combining methodological rigor with in-depth familiarity with PrEP, HIV prevention, and the lived realities of women sex workers. The full research team reviewed and discussed emerging findings throughout the analytic process. Finally, participant enrollment occurred within a fixed recruitment period rather than continuing until thematic saturation was reached, which may have limited the breadth of perspectives captured.

## Conclusions

Participants' deliberate decision to initiate and sustain PrEP adherence was based on a perceived need to reduce HIV risk from accidental or unconsented condomless sex. Participants reported PrEP adherence appeared to be sufficient to maintain HIV protection (more than four doses per week), and was facilitated by creation of daily routines, alarms, and personal motivation. Nevertheless, there were opportunities to boost adherence related to side effects from PrEP and its ingestion with other substances, and changes in daily routine related to work and travel. Intentional nonadherence related to side effects gained relevance due to the survival needs of this population. Secondary gains that reinforced PrEP adherence included a feeling of empowerment and the opportunity to opt for condom substitution for economic benefit, personal pleasure, or the desire to foster a trusting relationship with long-standing clients. This involved few selected encounters. PrEP communication between peers and with clients was limited by PrEP and HIV stigma.

In light of these results we believe our PrEP program can explore different avenues to increase uptake and adherence among women sex workers, from working on targeted PrEP awareness campaigns, to improving education on PrEP side effects and potential interactions with prescribed and recreational drugs, to enrolling peer-navigators, and creating strategies to facilitate PrEP dispensation. Moreover, the reported use of recreational drugs while engaging in sex work and a discrete decrease in condom use with selected clients highlights the need for PrEP programs to be a part of multifaceted sexual health clinics that offer harm reduction services and STI care.

## Supporting information

**S1 Table. Summary of descriptive quantitative findings at the individual level.**
(PDF)

**S2 Table. Joint display table with individual reports of PrEP use and side effects.**
(PDF)

**S3 Table. Joint display table with individual reports of condom use.**
(PDF)

**S4 Table. Joint display table with individual reports of number of clients.**
(PDF)

**S5 Table. Mixed analysis table for PrEP adherence, condom use and number of clients at the individual level.**
(PDF)

## Author contributions

**Conceptualization:** Laia Jimena Vazquez Guillamet, Jorge Valencia, Pablo Ryan, Mariano Matarranz, Guillermo Cuevas-Tascón, Miguel Angel del-Olmo-Morales, Jeffrey V Lazarus, Guillaume Chevance.

**Data curation:** Laia Jimena Vazquez Guillamet.

**Formal analysis:** Laia Jimena Vazquez Guillamet, Cristina Viladomiu, Lynn T Matthews, Guillaume Chevance.

**Funding acquisition:** Laia Jimena Vazquez Guillamet.

**Investigation:** Laia Jimena Vazquez Guillamet, Laura Laguna, Natalia Casanueva.

**Methodology:** Laia Jimena Vazquez Guillamet, Lynn T Matthews, Guillaume Chevance.

**Project administration:** Laia Jimena Vazquez Guillamet, Jorge Valencia, Guillaume Chevance.

**Resources:** Laia Jimena Vazquez Guillamet, Jorge Valencia, Pablo Ryan, Miguel Angel del-Olmo-Morales, Jeffrey V Lazarus.

**Software:** Laia Jimena Vazquez Guillamet, Guillaume Chevance.

**Supervision:** Laia Jimena Vazquez Guillamet, Jorge Valencia, Pablo Ryan, Lynn T Matthews, Jeffrey V Lazarus, Guillaume Chevance.

**Visualization:** Guillaume Chevance.

**Writing – original draft:** Laia Jimena Vazquez Guillamet, Guillaume Chevance.

**Writing – review & editing:** Laia Jimena Vazquez Guillamet, Jorge Valencia, Pablo Ryan, Mariano Matarranz, Guillermo Cuevas-Tascón, Miguel Angel del-Olmo-Morales, Laura Laguna, Natalia Casanueva, Cristina Viladomiu, Lynn T Matthews, Jeffrey V Lazarus, Guillaume Chevance.

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
