## [Decision Letter · Decision Letter 0]

10 Nov 2025

PGPH-D-25-02085

Intensive longitudinal follow-up of cisgender and transgender women engaged in sex work during the three months following PrEP initiation: a series of case-studies with mixed-method assessments

Dear Dr. Vazquez Guillamet,

Thank you for submitting your manuscript to PLOS Global Public Health. After careful consideration, we feel that it has merit but does not fully meet PLOS Global Public Health’s publication criteria as it currently stands. Therefore, we invite you to submit a revised version of the manuscript that addresses the points raised during the review process.

We look forward to receiving your revised manuscript.

Kind regards,

Jan Louis Adajo Non

Support Staff - Editorial

Journal Requirements:

i. State the initials, alongside each funding source, of each author to receive each grant. For example: "This work was supported by the National Institutes of Health (####### to AM; ###### to CJ) and the National Science Foundation (###### to AM)."

2. Please ensure that your Ethics Statement is available in its entirety at the beginning of your Methods section, under a subheading 'Ethics Statement'. It must include:

1) The name(s) of the Institutional Review Board(s) or Ethics Committee(s)

2) The approval number(s), or a statement that approval was granted by the named board(s)

3) (for human participants/donors) - A statement that formal consent was obtained (must state whether verbal/written) OR the reason consent was not obtained (e.g. anonymity).

Additional Editor Comments (if provided):

The authors are invited to revise the manuscript, and in particular, provide more details surrounding the methodology.

Reviewers' comments:

Reviewer's Responses to Questions

**Comments to the Author**

1. Does this manuscript meet PLOS Global Public Health’s publication criteria ? Is the manuscript technically sound, and do the data support the conclusions? The manuscript must describe methodologically and ethically rigorous research with conclusions that are appropriately drawn based on the data presented.

Reviewer #1: Yes

Reviewer #2: Yes

Reviewer #3: Yes

Reviewer #4: Yes

2. Has the statistical analysis been performed appropriately and rigorously?

Reviewer #1: Yes

Reviewer #2: Yes

Reviewer #3: I don't know

Reviewer #4: Yes

3. Have the authors made all data underlying the findings in their manuscript fully available (please refer to the Data Availability Statement at the start of the manuscript PDF file)?

Reviewer #1: Yes

Reviewer #2: Yes

Reviewer #3: Yes

Reviewer #4: Yes

4. Is the manuscript presented in an intelligible fashion and written in standard English?

Reviewer #1: No

Reviewer #2: Yes

Reviewer #3: Yes

Reviewer #4: Yes

5. Review Comments to the Author

Reviewer #1: Thank you for the opportunity to review the manuscript. This manuscript aims to understand the daily experiences of women sex workers using PrEP, with a focus on PrEP adherence. This is an important issue. Given substantial evidence on the effectiveness of PrEP, adherence is currently one of the main focuses of PrEP implementation. I commend the authors for their integrated mixed methods approach to tackle the research question. Below, I raise several points for consideration and/or clarification. In particular, the manuscript could benefit from more details on study methodology.

Major revisions

Introduction:

- Given that the focus of the paper is on women sex workers and the authors stratify results by transgender vs cisgender women sex workers, it would be helpful if the authors could provide more context on: 1) HIV prevalence/incidence among women sex workers rather than in the Spanish population overall [lines 57-58]; and 2) differences in social and structural contexts (e.g., stigma and discrimination, legal/structural barriers) and health risks (e.g., HIV prevalence/incidence, access to care, violence exposure, etc.) between transgender and cisgender women sex workers.

- Lines 60-61: The authors state that PrEP is subsidized. It would be useful for the authors to provide more details on subsidized access to PrEP. E.g., What proportion of PrEP costs is subsidized or are all PrEP costs covered? This information can help contextualize the role of cost on PrEP access/adherence.

Methods:

- More information on study setting/venue (e.g., was venue located in an urban, suburban, or rural region?) and how participants were recruited to the study (e.g., active recruitment via staff at the PrEP clinic or passive recruitment via posters/ads at the PrEP clinic) are needed. If available, it would also be interesting to see a breakdown of how many cisgender or transgender women presented at the PrEP clinic, how many were sex workers, how many were eligible for study participation, and, of these, how many accepted or declined to participate in the study. These numbers may speak to potential selection bias in the study and the representativeness of the study sample.

- Lines 114-116: It is unclear how the research team identified cisgender or TGW and sex workers? For instance, did participants need to self-report as a cisgender women or TGW sex worker or was there an existing database to draw from? It was also unclear how potential participants’ ability “to attend medical visits” was assessed at baseline? It would be helpful if these details could be included in the manuscript.

- It would be valuable for the authors to include a reflexivity of the research team detailing how interviewers' and analysts' backgrounds may have impacted data collected and the themes identified.

- It is unclear if the sample size was predetermined or if the authors kept recruiting until they reached saturation in the qualitative interviews? It would be helpful if these details could be included in the manuscript.

Discussion

- Lines 640-641: It is difficult to determine if the small sample included in the study was “representative” of the sex workers attending the clinic where the study took place as it is unclear how the research team identified sex workers at this clinic. For example, if sex workers were identified via self-report and numerous sex workers chose not to identify themselves as a sex worker, then the study sample would not be representative of sex workers attending the clinic. It may be advisable for the authors to reword their statement.

- The authors noted in the Results section that all participants were from Latin America. It may be helpful to add a few lines in the Discussion to consider the potential implications of this. E.g., how the cultural backgrounds of participants may influence their perceptions of PrEP, how they acquire PrEP knowledge, adherence, etc. It may also be helpful for authors to contextualize their findings in a landscape of expanding PrEP modalities where users may not need to take PrEP daily (e.g., event-driven PrEP, long-acting injectable PrEP).

Minor revisions

Abstract:

- Given the number of existing and emerging PrEP modalities, it would be helpful to specify/include “daily oral PrEP” in the title, abstract, and objective statement.

- It is unclear what the authors mean by “dynamics of PrEP adherence” in the abstract (lines 34-35) and introduction (line 108). Given this is a key outcome of the study, it would be helpful if the authors could define the term in the abstract and introduction.

- Line 38: The term “communication” is a bit vague. Specifying “communication between sex workers and their clients” would help improve clarity.

Introduction:

- Line 57-58: If data are available, stating how HIV incidence in Madrid compares to Spain and/or the rest of Europe would be valuable.

- Lines 76-78 state that, “adherence to PrEP with four or more doses per week is likely to provide optimal HIV protection for most individuals,[9] including cisgender women.[10]” Is this statement also true for transgender women?

- Lines 87-88: It would be helpful if these findings were among the sex work population or other populations.

- Line 96: It is unclear which behaviours the authors are referring to here as many have been mentioned so far.

Methods:

- There are inconsistency in recruitment dates: October 2022 to January 2023 (line 40) in Abstract vs November 8, 2022 and January 2023 in Methods (line 123-124) and October 2022 and January 2023 in Results (line 211). Please ensure recruitment dates are accurate and consistent across sections.

- Line 196: It would be helpful for authors to specify what they did for content analysis.

- Line 128: GDPR acronym was not defined.

Results:

- Line 339: The authors state that “When participants missed PrEP for one to three days in a row, it was usually due to side effects and/or work related issues.” It would be helpful to provide a statistic instead of using the word “usually”.

- Lines 348-349 state that, “The average was 32% and 65% for transgender and cisgender women sex workers, respectively (S1 Table).” It is unclear how 65% was calculated if there were only 3 cisgender FSW? It would be helpful if the authors double checked their %’s throughout the manuscript for accuracy.

Other notes:

- The manuscript could benefit from a spell check as there are several grammatical errors throughout (e.g., line 79 should state “over time” instead of “overtime”, line 133 should state “were” instead of “where”, line 185 should state “latter” instead of “later”, line 564 should state “coping” instead of “copying”, among others).

Reviewer #2: The manuscript is well-written, easy to follow, and demonstrates a clear understanding of the subject matter. The authors have provided adequate references to support their arguments, which enhances the credibility and scholarly depth of the paper. The structure is logical, and the flow of ideas allows readers to easily grasp the main objectives and findings of the study. However, there is room for improvement, particularly in ensuring that the language used is community-friendly and inclusive. The terminology should reflect the preferences and lived realities of the populations discussed. For instance, instead of using the term “woman sex workers,” the authors are encouraged to adopt “female sex workers,” which is more respectful and widely recognized within community and research contexts. Furthermore, it would be helpful for the authors to define and simplify technical terms such as “cisgender” to make the manuscript more accessible to diverse readers, including those without extensive academic or gender studies backgrounds. In addition, the paper would be strengthened by integrating a more comprehensive review of previous studies, particularly those focusing on similar populations or regional contexts. Highlighting past research will help situate the current study within the broader field, showing its contribution, gaps addressed, and relevance to ongoing discussions in community health and rights. Overall, the manuscript is promising and can make a valuable contribution once these adjustments are made.

Reviewer #3: Thank you for the opportunity to review this manuscript.

I suggest major revisions are necessary to the presentation of the findings prior to a more detailed review of the manuscript.

A mixed methods approach to the data collection and analysis is commendable, but the current presentation of findings by numerous themes followed by quantitative, qualitative, and mixed methods data is cumbersome and hard to follow. The data sets can be put into conversation and simply presented as mixed methods data under themes. Additionally, I would recommend the inclusion of fewer themes, with more of an analytic (and less descriptive) focus in order to more clearly present your main findings and key messages.

Additional points:

Introduction:

-Would be helpful to include more information on how PrEP is delivered and accessed in Spain, as well as information about the legal status of sex work.

-As there is such a vast body of literature on PrEP, please consider adding additional references to the statements in the Introduction section.

-Line 70: delete orientation (can just say gender)

-Line 76: delete ‘the’ in front of ‘PrEP’

-Consider avoiding acronyms, such as TWG

Methods and Materials

-Can you further describe what an initial PrEP visit entails?

-How was involvement in sex work known? Especially as sex workers often do not disclose their occupation in clinical settings due to stigma.

-As the sample size is quite small, especially considering 3 months of enrollment, further details are required about the enrollment process.

-Consider moving some of the supplementary materials (i.e. screenshot of the app) to the main document. Can you also provide more details on the app? How was it developed? Has it been used before?

-The Medication Adherence Model needs to be described within the text.

Results

-Table 1 can be presented as aggregate data and not per participant.

-Line 270-1: “When considering only the days with answers to the daily survey, adherence oscillated between 77.7% to 100% (95% in average for both groups) (S1 Table).” I am not clear on why you would or could exclude days without responses – wouldn’t a lack of response be anticipated to be linked with lower adherence?

-The findings section is very long. It is not clear what the Mixed Methods section is adding. Instead, the quantitative and qualitative findings need to be put into conversation and thematically presented together. Additional supplementary tables might not be needed if key findings are presented in the body of the manuscript. Much of the data presented in the findings is already known and well-documented in the literature. I would encourage the authors to focus in on which data points are novel. Further analysis can be done as many of the themes can be lumped together and presented with more analytic detail.

-I would not include a subsection with just two cisgender women sex workers.

-Please note that I am not able to assess the heat maps or models.

Discussion

-Subheadings do not need to be repeated in the discussion.

-Line 569 – I would avoid saying sex work negatively impacts PrEP adherence, but rather shift work and changing schedules/routines.

-Line 573-4: “In addition, the use of alcohol and drugs with clients impacted PrEP intake, sometimes because it made them more forgetful, but mostly because participants opt to skip PrEP to avoid unintended side effects of mixing” – was this presented in the data?

-Line 583-4: “We hypothesize that stigma or forgetfulness prevented them from sharing these concerns with their prescribing physicians.”-is there anything in the data pointing to forgetfulness related to inquiring about medication interactions? Discussion statements should link to what was presented in the findings section (and not in the supplementary tables).

-Line 595-7: “The report of the cisgender women in our study is in concordance with the lack of risk compensating behaviors identified in the only study on this area carried amongst female sex workers in Benin.[41]” – I believe many PrEP studies have also looked at condom use. This statement should be clarified.

Limitations

-I am not sure that you can call your sample representative.

-Limitations require more discussion of the enrolment process, small sample size, and need for disclosure.

-“ The fact that cisgender women sex workers’ need to disclose sex work status to qualify for PrEP, along with lack of awareness, may explain the disproportionate ratio between transgender and cisgender women sex workers in the study” -but transgender women would also need to disclose sex work.

-Conclusion can be shortened and the Figure should not be in the conclusion section. It’s also not clear how the Medication Adherence Model was applied in the study – this should have been central to the presentation of findings.

Reviewer #4: Yes from my quick search this has not been published before. One article was published on determinants of intention to use PrEP (https://link.springer.com/article/10.1007/s10508-024-02834-4). See my feedback/comments below for minor revisions.

Additional Comments

- In introduction, would change per "100.000 habitants to 100,000" same for "114.576" in Introduction

- Line 61-62 would specify dosing at least once of tenofovir disoproxil fumarate/emtricitabine 300/200 mg po daily and/or other regimens as appropriate)

- Line 91-93 can you clarify what the Benin paper showed, not clear

- Line 116 methods would clarify that you mean does not have chronic hep B infection

- Line 128-129 what is GDPR-compliant m-path application (wording also to be checked -- you say application twice in sentence)

- Results: Demographics line 209 change to "approached for recruitment for the study."

- Line 240 spelling of penis to be corrected

- Line 267 can you report the PreP adherence by which participant (ID 1 through 15) ie which participant had adherence of 37% and which had 100% (if you can report)

- Can shorten the discussion where you resummarize the results (would synthesize more tightly)

- Figure 2 clarify x axis

- Overall can shorten # of quotes in body of text since you have quotes in the supplementary files

6. PLOS authors have the option to publish the peer review history of their article (what does this mean? ). If published, this will include your full peer review and any attached files.

**Do you want your identity to be public for this peer review?** For information about this choice, including consent withdrawal, please see our Privacy Policy .

Reviewer #1: **Yes:** Min Xi

Reviewer #2: No

Reviewer #3: No

Reviewer #4: No

Figure Resubmissions:

---

## [Editor Report · Decision Letter 1]

9 Feb 2026

Intensive longitudinal follow-up of cisgender and transgender women engaged in sex work during the three months following initiation of daily oral PrEP: a series of case-studies with mixed-method assessments

PGPH-D-25-02085R1

Dear Dr Vazquez Guillamet,

We are pleased to inform you that your manuscript 'Intensive longitudinal follow-up of cisgender and transgender women engaged in sex work during the three months following initiation of daily oral PrEP: a series of case-studies with mixed-method assessments' has been provisionally accepted for publication in PLOS Global Public Health.

Best regards,

Sharmistha Mishra, M.D., Ph.D

Academic Editor

The authors have carefully addressed and responded to the peer review comments.